Alcohol consumption and risk of fatty liver disease: a meta-analysis

Cao Guoli 1
Yi Tingzhuang 1 2
Liu Qianqian 1
Wang Min 1
Tang Shaohui tangshaohui206@163.com 1
1 Department of Gastroenterology, The First Affiliated Hospital, Jinan University , Guangzhou , Guangdong , China
2 Department of Gastroenterology, Affiliated Hospital of Youjiang Medical University for Nationlities , BaiSe , GuangXi , China
Patton Bob
Electronic publication date: 2016 Oct 27
Publication date: 2016
Volume: 4
Electronic Location ID: e2633
Received 2016 Aug 9; Accepted 2016 Sep 27
Copyright: ©2016 Cao et al.
Copyright year: 2016
Copyright holder: Cao et al.
License: This is an open access article distributed under the terms of the Creative Commons Attribution License, which permits unrestricted use, distribution, reproduction and adaptation in any medium and for any purpose provided that it is properly attributed. For attribution, the original author(s), title, publication source (PeerJ) and either DOI or URL of the article must be cited.
License URL: https://creativecommons.org/licenses/by/4.0/

Keywords: Alcohol, Fatty liver disease, Risk, Meta-analysis

Funding: The authors received no funding for this work.

==============================
Background

Observational studies have shown inconsistent results regarding alcohol consumption and risk of fatty liver. We performed a meta-analysis of published literature to investigate the association between alcohol consumption and fatty liver disease (FLD).

Methods

We searched Medline, Embase, Web of Science, and several Chinese databases, identifying studies that reported an association between alcohol consumption and the risk of FLD.

Results

A total of 16 studies with 76,608 participants including 13 cross-sectional studies, two cross-sectional following longitudinal studies, and one cohort study met the inclusion criteria. For light to moderate alcohol consumption (LMAC), there was a 22.6% reduction in risk of FLD (odds ratio [OR] = 0.774, 95% confidence interval CI [0.695–0.862], P <0.001), and subgroup analysis showed that a greater reduction in risk of FLD was found in the female drinkers (30.2%) and the drinkers with BMI ≥25 kg/m2(31.3%) compared with the male drinkers (22.6%) and the drinkers with BMI <25 kg/m2(21.3%), respectively. For heavy alcohol consumption, there was no significant influence on risk of FLD (OR = 0.869, 95% CI [0.553–1.364], P = 0.541) in Japanese women, but there was a 33.7% reduction in risk of FLD (OR = 0.663, 95% CI [0.574–0.765], P < 0.001) in Japanese men and a significant increased risk of FLD (OR = 1.785, 95% CI [1.064–2.996], P = 0.028) in Germans.

Conclusion

LMAC is associated with a significant protective effect on FLD in the studied population, especially in the women and obese population. However, the effect of heavy alcohol consumption on FLD remains unclear due to limited studies and small sample sizes.

Introduction

Fatty liver disease (FLD) is caused by the excessive accumulation of fat in the liver cells (Bedogni, Nobili & Tiribelli, 2014) which encompasses a morphological spectrum consisting of hepatic steatosis (fatty liver) and steatohepatitis that can progress to cirrhosis and hepatocellular carcinoma (Reddy & Rao, 2006). FLD is commonly divided into alcoholic liver disease (ALD) and nonalcoholic fatty liver disease (NAFLD) (Reddy & Rao, 2006).

ALD is a liver injury as a consequence of excessive or harmful alcohol use, which includes a spectrum of injury, ranging from simple steatosis to cirrhosis (O’Shea, Dasarathy & McCullough, 2010; Torruellas, French & Medici, 2014). NAFLD is defined as either the excessive fat accumulation or steatosis in the liver in patients who consume less than or equal to 30 g of alcohol per day for men and 20 g of alcohol per day for women after the exclusion of other causes such as hepatitis virus infection, use of steatogenic medication or hereditary disorders (Abd El-Kader & El-Den Ashmawy, 2015). NAFLD has been considered to be the hepatic manifestation in the patients with metabolic syndrome (Angulo et al., 1999), but it may also occur in 29% of lean patients lacking associative risk factors (Bugianesi et al., 2005).

Although it has long been known that long-term heavy drinking is a cause of liver cirrhosis and liver cancer, the findings from recent observational studies have shown that light (Dunn, Xu & Schwimmer, 2008; Nishioji et al., 2015), moderate (Moriya et al., 2013), and even heavier alcohol consumption (Gunji et al., 2009; Moriya et al., 2015) may decrease the risk of FLD. The mechanisms that explain the inverse association between alcohol consumption and FLD risk remain unknown, and the suggested mechanisms of protection by alcohol consumption include decreased insulin resistance, enhanced hepatic blood flow, antioxidant agents in alcoholic beverages, decreased triglyceride content in the liver, and increased circulating adiponectin (Moriya et al., 2013), Conversely, Lau et al. (2015) indicated that light alcohol consumption was associated with a higher prevalence of FLD; Cotrim et al. (2009) reported that light to moderate alcohol consumption (LMAC) had no impact on the severity of activity and stage of FLD. Since the effect of alcohol consumption on FLD development is still controversial, we therefore combined all published epidemiologic studies on this issue to evaluate the association between alcohol consumption and FLD risk.

Materials and Methods

Search strategy

Guoli Cao and Tingzhuang Yi independently searched Medline, Embase, Web of Science, and Chinese data sources including CNKI, Wanfang Data, and the VIP database without year restrictions, identifying studies that reported an association between alcohol consumption and the risk of FLD. Key words searched were as follows: (“alcohol” OR “alcohols” OR “ethanol” OR “drinking” OR “wine” OR “beer” OR “spirits” OR “prevalence”) AND (“fatty liver” OR “hepatic steatosis” OR “steatohepatitis”). We also checked the reference lists of the articles retrieved from PubMed search. English and Chinese language was used. Two independent reviewers made an initial judgment of whether the studies were eligible to be included in the meta-analysis, and any disagreements were resolved by consulting Shaohui Tang.

Inclusion and exclusion criteria

The inclusion criteria required studies to: (1) have cross-sectional, case–control, cohort study or randomized controlled trial (RCT) design; (2) provide information on alcohol consumption in relation to FLD, and the referent group are non-drinkers; (3) report odds ratios (ORs) with their corresponding 95% confidence intervals (CIs) or original data allowing us to compute them; (4) diagnose populations with fatty liver (hepatic steatosis) or steatohepatitis through the imaging, laboratory tests or liver biopsy. Exclusion criteria included duplicate reports, abstracts, case reports, review articles, editorials, and clinical guidelines.

Data extraction

Data extracted from each study included the name of the first author, study design, study region, study period, publication year, diagnostic method, the age and sex of subjects, sample size, adjustments, inclusion and exclusion criteria, average consumption of alcohol, the OR and their 95% CI. All risk estimates are converted to OR by directly extracting from the study or calculating from raw data. The data extraction was performed by Guoli Cao and Qianqian Liu. Investigators independently reviewed and cross-checked the data, and any disagreements were resolved by discussion between two authors or by consulting Shaohui Tang. If results were published more than once, the results from the most recent one were selected.

Definition of different alcohol consumption levels

According to the recommendations of the World Health Organization (WHO, 2000), the average ethanol intake per drinking day can be usefully classified as “Low risk” (≤20 g/day for women and ≤40 g/day for men), “Medium risk” (>20–40 g/day for women and >40–60 g/day for men) and “High risk” (>40 g/day for women and >60 g/day for men). Based on the different levels of alcohol consumption reported in the included studies, we classified the drinkers into four groups: non-drinkers, persons who drink 0 g/day of alcohol; light drinkers, persons who drink ≤20 g/day (or ≤140 g/week) of alcohol; moderate drinkers, persons who drink >20–40 g/day (or >140–280 g/week); and heavy drinkers, persons who drink >40 g/day (or >280 g/week).

Quality assessment

The quality of cross-sectional studies was assessed by the Agency for Healthcare Research and Quality (AHRQ) criteria (Rostom et al., 2004). The quality of cohort study was assessed by Newcastle-Ottawa Scale (NOS) (Wells et al., 2011), including representativeness of the exposed cohort, selection of the unexposed cohort, ascertainment of exposure, outcome of interest not present at start of study, control for the most important factor or the second important factor, outcome assessment, follow-up long enough for outcomes to occur, adequacy of follow-up of cohorts.

Statistical analysis

Statistical analysis was performed using STATA version 12.0 (Stata Corp, College Station, Texas). The results were expressed in terms of OR and 95% CI. Statistical heterogeneity was evaluated through the Q test and I2 statistic (Higgins & Thompson, 2002; Higgins et al., 2003), and P < 0.05 was considered statistically significant. The test statistic was distributed as χ2. Q statistics was used to evaluate heterogeneity, with its P values revealed by the forest plot. I2 was used to estimate the size of the heterogeneity, with its P values revealed by the forest plot. If the heterogeneity was acceptable (I2 < 50%), a fixed-effects model was conducted to calculate the pooled OR. Conversely, a random-effects model was used. The causes of heterogeneity were investigated by subgroup analysis. To evaluate whether the pooled results might be influenced by individual studies, a sensitivity analysis was performed by omitting one study each time and recalculating the pooled OR. We applied Egger’s test and Begg’s method to assess bias. A two-sided P value of less than 0.05 was regarded as significant.

Figure 1 Search strategy and flow of information relative to the meta-analysis.

Results

Search results and study characteristics

Figure 1 shows the process of selecting studies for the meta-analysis. The searches yield 2,847 studies from Chinese and English databases, and 2,622 studies were excluded based on title. Among the remaining 225 studies, 202 were further excluded based on abstract or full text because they did not fulfill the inclusion criteria. Then, 19 English and four Chinese studies remained for complete evaluation, four Chinese studies were excluded due to design weakness or low quality data, two English studies were excluded due to a lack of data for calculation, and one English study was discarded due to duplication. In the end, 16 observational articles with a total of 76,608 participants including 39,198 nondrinkers, 31,942 light to moderate drinkers (LM drinkers), and 5,468 heavy drinkers met our inclusion criteria (Dunn, Xu & Schwimmer, 2008; Cotrim et al., 2009; Gunji et al., 2009; Yamada et al., 2010; Hiramine et al., 2011; Dunn et al., 2012; Hamaguchi et al., 2012; Wong et al., 2012; Moriya et al., 2013; Sookoian, Castaño & Pirola, 2014; Hashimoto et al., 2015; Kächele et al., 2015; Lau et al., 2015; Moriya et al., 2015; Nishioji et al., 2015; Sogabe et al., 2015). There were 13 cross-sectional studies (Dunn, Xu & Schwimmer, 2008; Cotrim et al., 2009; Gunji et al., 2009; Hiramine et al., 2011; Dunn et al., 2012; Hamaguchi et al., 2012; Wong et al., 2012; Moriya et al., 2013; Sookoian, Castaño & Pirola, 2014; Kächele et al., 2015; Lau et al., 2015; Nishioji et al., 2015; Sogabe et al., 2015), two cross-sectional following longitudinal studies (Yamada et al., 2010; Moriya et al., 2015) and one cohort study (Hashimoto et al., 2015). Of the studies, 10 were conducted in Asia (nine in Japan (Gunji et al., 2009; Yamada et al., 2010; Hiramine et al., 2011; Hamaguchi et al., 2012; Moriya et al., 2013; Hashimoto et al., 2015; Moriya et al., 2015; Nishioji et al., 2015; Sogabe et al., 2015) and one in Hong Kong (Wong et al., 2012)) and six in other countries (two in the US (Dunn, Xu & Schwimmer, 2008; Dunn et al., 2012), 1 in Brazil (Cotrim et al., 2009), 1 in Argentina (Sookoian, Castaño & Pirola, 2014) and 2 in Germany (Kächele et al., 2015; Lau et al., 2015)). A total of 12 studies provided adjusted risk estimate, and four studies reported only crude data. One study (Moriya et al., 2011) was excluded because it was duplicate study (Tables 1 and 2).

Table 1 Characteristics of studies included in the meta-analysis.

Study	Design	Study region	Study period	Outcome	Diagnostic method	BMI (mean)	Age(years)	
Dunn, Xu & Schwimmer, 2008	Cross-section	United States	1988–1994	FLD	Laboratory examination	26.99	>21	
Cotrim et al., 2009	Cross-section	Brazil	2004–2005	FLD	Biopsy	43.9	37.27 ± 11.06	
Gunji et al., 2009	Cross-section	Japan	2007–2008	FLD	Ultrasound	23.5	50.9 ± 8.1a	
Yamada et al., 2010	Cross-sectional and longitudinal	Japan	2000–2005	FLD	Ultrasound	22.58	49.8 ± 10.7a 50.4 ± 9.3b	
Hiramine et al., 2011	Cross-section	Japan	2000–2007	FLD	Ultrasound	23.7	30–69a	
Wong et al., 2012	Cross-section	Hong Kong	2008–2010	FLD	Ultrasound	22.8	48 ± 11	
Dunn et al., 2012	Cross-section	United States	–	FLD	Biopsy	34.36	>21	
Hamaguchi et al., 2012	Cross-section	Japan	2004–2009	FLD	Ultrasound	22.57	18–88	
Moriya et al., 2013	Cross-section	Japan	2003–2006	FLD	Ultrasound	21.8	46.4 ± 8.9b	
Sookoian, Castaño & Pirola, 2014	Cross-section	Argentina	–	FLD	Laboratory examination and biopsy	29.82	–	
Hashimoto et al., 2015	Retrospective cohort	Japan	1994–2003	FLD	Ultrasound	22.25	–	
Kächele et al., 2015	Cross-section	Germany	–	FLD	Ultrasound	25.81	18–49	
Moriya et al., 2015	Cross-sectional and longitudinal	Japan	2004–2006	FLD	Ultrasound	23.04	49.1 ± 8.3a 47.6 ± 8.1b	
Sogabe et al., 2015	Cross-section	Japan	2008–2012	FLD	Ultrasound	27.0	21–81a	
Lau et al., 2015	Cross-section	Germany	–	FLD	Ultrasound	26.73	32–69	
Nishioji et al., 2015	Cross-section	Japan	2011–2012	FLD	Ultrasound	27.32a 20.52b	22–92	
Notes.

a Male data.

b Female data.

BMI body mass index

FLD fatty liver disease

Table 2 Summary of the results of studies included in the meta-analysis.

Study	Male/ Female	Alcohol consumption	OR (95% CI)	Covariate adjustments	
	Non drinkers	LM drinkers	Heavy drinkers				
Dunn, Xu & Schwimmer, 2008	2,553/4,658	2,315/2,228	0/0	Nondrinkers	1	Age, gender, race, income, education, neighborhood population density, caffeine consumption, physical activity	
			10 g/day	0.70(0.53, 0.93)	
Cotrim et al., 2009	14/43	27/48	0/0	Nondrinkers	1	None	
			≤20g/day	0.98(0.30, 3.24)	
			>20–40 g/day	1.0(0.18, 5.43)	
Gunji et al., 2009	1,706/0	2,879/0	1,014/0	Nondrinkers	1	Age, body mass index, waist girth, visceral adipose tissue, subcutaneous adipose tissues, systolic blood pressure, diastolic blood pressure, high-density lipoprotein, cholesterol, low-density lipoprotein cholesterol, triglycerides, fasting blood glucose, glycated hemoglobin alanine aminotransferase, smoking status, dietary habits, physical activity	
			40–140 g/week	0.82(0.68, 0.99)a	
			>140–280 g/week	0.75(0.61, 0.93)a	
			>280 g/week	0.85(0.67, 1.09)a	
Yamada et al., 2010	1,040/3,063	3,476/1,857	928/60	Nondrinkers	1	Age, body mass index, smoking status	
			Occasional	0.95(0.77, 1.17)a	
			drinkers	0.81(0.63, 1.04)b	
			23 g/day	0.72(0.58, 0.89)a 0.71(0.44, 1.16)b	
			>46 g/day	0.65(0.50, 0.85)a 0.74(0.25, 2.17)b	
Hiramine et al., 2011	847/0	4,540/0	347/0	Nondrinkers	1	Age, body mass index, alanine aminotransferase, aspartate aminotransaminase, γ-glutamyl transpeptidase, triglycerides, high-density lipoprotein	
			<20 g/day	0.71(0.59, 0.86)a	
			>60 g/day	0.44(0.32, 0.62)a	
Wong et al., 2012	720c	148c	0c	Nondrinkers	1	Demographic, metabolic factors	
			<10 g/day	1.37(0.89, 2.11)	
Dunn et al., 2012	70/181	128/203	0/0	Non-drinks	1	Gender, age, race, income, education, glycated hemoglobin, alanine aminotransferase, recreational, non-recreational physical activity, smoking, total calories per day, percent calories from carbohydrates, percent calories from fat	
			<20 g/day	0.56(0.39, 0.84)	
Hamaguchi et al., 2012	6,154/6,892	3,350/613	1,478/84	Nondrinkers	1	Age, use of drugs. metabolic syndrome, regular exercise, smoking	
			40–140 g/week	0.69(0.60, 0.79)a 0.54(0.34, 0.88)b	
			>140–280 g/week	0.72(0.63, 0.83)a 0.43(0.21, 0.88)b	
			>280 g/week	0.74(0.64, 0.85)a 1.02(0.44, 2.35)b	
Moriya et al., 2013	0/3,403	0/1,219	0/0	Nondrinkers	1	Obesity, atherogenic, dyslipidemia, glucose intolerance, hyperuricemia, hypertension, current smoking status, age	
			<70 g/week	0.74(0.55, 0.98)b	
			70–139.9 g/week	0.67(0.44, 1.00)b	
Sookoian, Castaño & Pirola, 2014	172/159	40/43	0/0	Nondrinkers	1	–	
			<40 g/day	0.49(0.30, 0.79)	
Hashimoto et al., 2015	1,704/1,765	1,332/208	411/17	Nondrinkers	1	None	
			40–140 g/week	0.602(0.486 ,0.745)a 0.539(0.216,1.344)b	
			>140–280 g/week	0.607(0.484, 0.763)a 0.366(0.050, 2.680)b	
			>280 g/week	0.573(0.436, 0.751)a 1.052(0.138 ,8.012)b	
Kächele et al., 2015	33/67	114/86	118/14	Nondrinkers	1	None	
			≤20g/day	0.44(0.23, 0.83)	
			>20–40 g/day	0.96(0.53, 1.71)	
			>40 g/day	1.29(0.75, 2.20)	
Moriya et al., 2015	971/1,088	2047/420	755/16	Nondrinkers	1	Obesity, regular exercise, smoking	
			<70g/week	0.71(0.52, 0.96)a 0.79(0.68, 0.90)b	
			70–139.9 g/week	0.73(0.63, 0.84)a 0.67(0.45, 0.98)b	
			>140–280 g/week	0.69(0.60, 0.79)a 0.86(0.54, 1.37)b	
			>280g/week	0.68(0.58, 0.79)a 0.82(0.43, 1.57)b	
Sogabe et al., 2015	281/0	774/0	0/0	Nondrinkers	1	Age, body mass index, waist circumference, hypertension, dyslipidaemia, uric acid, glycated hemoglobin, alanine aminotransferase, metabolic syndrome type	
			drinking <20 g/day	0.65(0.47, 0.91)a	
Lau et al., 2015	184/250	1,460/1,638	213/13	Nondrinkers	1	Age, body mass index, glycated hemoglobin alanine aminotransferase, menopausal status in female	
			≤10g/day	1.19(0.85, 1.66)a 0.67(0.46, 0.98)b	
			>10–20 g/day	1.53(1.15, 2.05)a 0.65(0.43, 0.98)b	
			>20–40 g/day	2.03(1.51, 2.72)a 0.65(0.34, 1.23)b	
			>40–60 g/day	2.18(1.61, 2.94)a	
			>60–80 g/day	2.24(1.62, 3.10)a	
Nishioji et al., 2015	165/1015	175/574	0/0	Nondrinkers	1	Age, body fat percentage, body mass index, waist circumference, diastolic blood pressure, total protein, serum albumin, alanine aminotransferase, cholinesteras, triglycerides, high-density lipoprotein, glycated hemoglobin, blood pressure risk	
			<20 g/day	0.49(0.27, 0.90)a 0.66(0.44, 0.99)b	
Notes.

a Male data.

b Female data.

c Total number.

LM drinkers light to moderate drinkers

OR odds ratio

CI confidence interval.

Study quality

Most of the cross-sectional studies had provided specific inclusion and exclusion criteria, source of information, and controlled confounding factors. But only a few studies obtained a follow-up (Yamada et al., 2010; Moriya et al., 2015) and explained how missing data were handled (Dunn, Xu & Schwimmer, 2008; Moriya et al., 2013; Kächele et al., 2015; Lau et al., 2015; Moriya et al., 2015; Nishioji et al., 2015; Sogabe et al., 2015). Most of the studies (Dunn, Xu & Schwimmer, 2008; Gunji et al., 2009; Hiramine et al., 2011; Hamaguchi et al., 2012; Wong et al., 2012; Moriya et al., 2013; Kächele et al., 2015; Lau et al., 2015; Nishioji et al., 2015; Sogabe et al., 2015; Yamada et al., 2010; Moriya et al., 2015) were evaluated as low risk of bias, 2 studies (Cotrim et al., 2009; Dunn et al., 2012) had moderate risk of bias, and 1 study (Sookoian, Castaño & Pirola, 2014) had high risk of bias (Table 3). The quality of cohort study (Hashimoto et al., 2015) was full score of 9, and a score ≥6 stars is considered to be high quality.

Table 3 Quality assessment of cross-sectional studies included in this meta- analysis.

Study	Item 1	Item 2	Item 3	Item 4	Item 5	Item 6	Item 7	Item 8	Item 9	Item 10	Item 11	
Dunn, Xu & Schwimmer, 2008	Y	Y	Y	Y	U	Y	Y	Y	Y	Y	N	
Cotrim et al., 2009	Y	Y	Y	Y	U	Y	N	N	N	Y	N	
Gunji et al., 2009	Y	Y	Y	Y	N	Y	N	Y	N	Y	N	
Yamada et al., 2010	Y	Y	Y	Y	N	Y	N	Y	N	Y	Y	
Hiramine et al., 2011	Y	Y	Y	Y	N	Y	N	Y	N	Y	N	
Wong et al., 2012	Y	Y	Y	Y	N	Y	Y	Y	N	Y	N	
Dunn et al., 2012	Y	Y	N	U	U	Y	Y	Y	N	Y	N	
Hamaguchi et al., 2012	Y	Y	Y	Y	N	Y	N	Y	N	Y	N	
Moriya et al., 2013	Y	N	Y	Y	N	Y	Y	Y	Y	Y	N	
Sookoian, Castaño & Pirola, 2014	Y	U	U	Y	N	Y	U	U	U	Y	N	
Kächele et al., 2015	Y	Y	N	Y	N	Y	Y	N	Y	Y	N	
Moriya et al., 2015	Y	N	Y	Y	N	Y	Y	Y	Y	Y	Y	
Sogabe et al., 2015	Y	Y	Y	Y	U	Y	Y	Y	Y	Y	N	
Lau et al., 2015	Y	Y	N	Y	N	Y	Y	Y	Y	Y	N	
Nishioji et al., 2015	Y	Y	Y	Y	N	Y	Y	Y	Y	Y	N	
Notes.

Y, yes; N, no; U, unclear; Item 1, define the source of information (survey, record review); Item 2, list inclusion and exclusion criteria for exposed and unexposed subjects (cases and controls) or refer to previous publications; Item 3, indicate time period used for identifying patients; Item 4, indicate whether or not subjects were consecutive if not population-based; Item 5, indicate if evaluators of subjective components of study were masked to other aspects of the status of the participants; Item 6, describe any assessments undertaken for quality assurance purposes (e.g., test/retest of primary outcome measurements); Item 7, explain any patient exclusions from analysis; Item 8, describe how confounding was assessed and/or controlled; Item 9, if applicable, explain how missing data were handled in the analysis; Item 10, summarize patient response rates and completeness of data collection; Item 11, clarify what follow-up, if any, was expected and the percentage of patients for which incomplete data or follow-up was obtained.

Light to moderate drinkers (LM drinkers) vs non-drinkers

A meta-analysis was conducted with the data from the 16 heterogeneous studies (I2 = 79.3%, P < 0.001) with 31,942 LM drinkers, showing the LMAC was associated with a 22.6% reduction in risk of FLD (OR = 0.774, 95% CI [0.695–0.862], P < 0.001) using random effect model (Fig. 2). Subsequently, we conducted a sensitivity analysis by omitting one study each time and recalculating the pooled OR, and the results showed that the pooled risk estimates did not change significantly. There was a symmetric funnel plot and no evidence of significant publication bias from Egger’s test (P = 0.969) and Begg’s test (P = 0.753) of the 16 studies.

Then, when a stratified analysis was conducted according to different amounts of alcohol consumption, a total of 15 heterogeneous studies (Dunn, Xu & Schwimmer, 2008; Cotrim et al., 2009; Gunji et al., 2009; Yamada et al., 2010; Hiramine et al., 2011; Dunn et al., 2012; Hamaguchi et al., 2012; Wong et al., 2012; Moriya et al., 2013; Hashimoto et al., 2015; Kächele et al., 2015; Lau et al., 2015; Moriya et al., 2015; Nishioji et al., 2015; Sogabe et al., 2015) were included in light alcohol consumption group (I2 = 66%; P < 0.001), and eight heterogeneous studies (Cotrim et al., 2009; Gunji et al., 2009; Yamada et al., 2010; Hamaguchi et al., 2012; Hashimoto et al., 2015; Kächele et al., 2015; Lau et al., 2015; Moriya et al., 2015) in moderate alcohol consumption group (I2 = 82.7%; P < 0.001). The combined analysis showed a greater reduction (25.3%) in risk of FLD in light alcohol consumption group (OR = 0.747; 95% CI [0.673–0.830]; P < 0.001) compared with moderate alcohol consumption group (19.6%) (OR = 0.804; 95% CI [0.661–0.979]; P = 0.03) in random effect model (Fig. 3).

Figure 2 Forest plot for assessing the association between light to moderate alcohol consumption and FLD.

Figure 3 Forest plot for assessing the association between different amounts of alcohol consumption and FLD.

Further, a subgroup analysis was conducted by sex. In women, the result showed that LMAC was associated with a 30.2% reduction in risk of FLD (OR = 0.698, 95% CI [0.628–0.776], P < 0.001) using the 7 studies with 5,955 LM drinkers (Yamada et al., 2010; Hamaguchi et al., 2012; Moriya et al., 2013; Hashimoto et al., 2015; Lau et al., 2015; Moriya et al., 2015; Nishioji et al., 2015) without significant heterogeneity (I2 = 0.0%, P = 0.571) with fixed effect model. In men, the 9 heterogeneous studies (Gunji et al., 2009; Yamada et al., 2010; Hiramine et al., 2011; Hamaguchi et al., 2012; Hashimoto et al., 2015; Lau et al., 2015; Moriya et al., 2015; Nishioji et al., 2015; Sogabe et al., 2015) (I2 = 90.2%, P < 0.001) with 19,858 LM drinkers were included in the analysis, only showing a 22.6% reduction in risk of FLD (OR = 0.774, 95% CI [0.657–0.913], P = 0.002) in relation to the LMAC using random effect model (Fig. 4). Sensitive analysis indicated that no individual studies could change the pooled results in women and in men.

Figure 4 Forest plot of subgroup analysis conducted by sex for assessing the association between light to moderate alcohol consumption and FLD.

Finally, we also conducted another subgroup analysis by BMI (body mass index). In the groups with BMI ≥25 kg/m2 and <25 kg/m2 for subjects, there were 8 heterogeneous (I2 = 82.0%, P < 0.001) (Dunn, Xu & Schwimmer, 2008; Cotrim et al., 2009; Dunn et al., 2012; Sookoian, Castaño & Pirola, 2014; Kächele et al., 2015; Lau et al., 2015; Nishioji et al., 2015; Sogabe et al., 2015) and nine heterogeneous (I2 = 70.8%, P < 0.001)(Gunji et al., 2009; Yamada et al., 2010; Hiramine et al., 2011; Hamaguchi et al., 2012; Wong et al., 2012; Moriya et al., 2013; Hashimoto et al., 2015; Moriya et al., 2015; Nishioji et al., 2015) studies were included, respectively. The combined results showed a greater protective effect of LMAC on FLD development in the drinkers with BMI ≥25 kg/m2(OR = 0.687, 95% CI [0.508–0.930], P = 0.015) compared with the drinkers with BMI <25 kg/m2 (OR = 0.787, 95% CI [0.715–0.866], P < 0.001) using random effect model (Fig. 5).

Figure 5 Forest plot of subgroup analysis conducted by BMI for assessing the association between light to moderate alcohol consumption and FLD.

Heavy drinkers vs non-drinkers

Significant heterogeneity was found among the eight studies (six conducted in Japan and two in Germany) with 5,468 heavy drinkers (Gunji et al., 2009; Yamada et al., 2010; Hiramine et al., 2011; Hamaguchi et al., 2012; Hashimoto et al., 2015; Kächele et al., 2015; Lau et al., 2015; Moriya et al., 2015) (I2 = 93.6%, P < 0.001), and there was no difference in risk of FLD between heavy drinkers and nondrinkers (OR = 0.815, 95 %CI [0.59–41.120], P = 0.208) using random effect model (Fig. 6). There was a symmetric funnel plot and no evidence of significant publication bias from Egger’s test (P = 0.868) and Begg’s test (P = 0.536) of the 8 studies.

Owing to significant heterogeneity, the above 8 studies were divided into group A (the six studies from Japan (Gunji et al., 2009; Yamada et al., 2010; Hiramine et al., 2011; Hamaguchi et al., 2012; Hashimoto et al., 2015; Moriya et al., 2015)) and group B (the 2 studies from Germany (Kächele et al., 2015; Lau et al., 2015)). In the group A, the result showed that heavy alcohol consumption was associated with a 33.2% reduction in risk of FLD (OR = 0.668, 95% CI [0.579–0.770], P < 0.001) with decreased heterogeneity (I2 = 61.9%, P < 0.001) using random effect model (Fig. 6). Subgroup analysis conducted by sex indicated that there was no difference in risk of FLD between heavy drinkers and nondrinkers in women (OR = 0.869, 95% CI [0.553–1.364], P = 0.541) using the four studies (Yamada et al., 2010; Hamaguchi et al., 2012; Hashimoto et al., 2015; Moriya et al., 2015) without significant heterogeneity (I2 = 0%, P = 0.962) with fixed effect model, but in men a 33.7% reduction (OR = 0.663, 95% CI [0.574–0.765], P < 0.001) was found in risk of FLD regarding heavy alcohol consumption using the six studies (Gunji et al., 2009; Yamada et al., 2010; Hiramine et al., 2011; Hamaguchi et al., 2012; Hashimoto et al., 2015; Moriya et al., 2015) with decreased heterogeneity (I2 = 61.6%, P = 0.023) using random effect model (Fig. 7). In the group B, 132 heavy drinkers and 213 male heavy drinkers were included by Kächele et al. (2015) and Lau et al. (2015), respectively, and an increased risk of FLD was observed in relation to heavy alcohol consumption (OR = 1.785, 95% CI [1.064–2.996], P = 0.028) with decreased heterogeneity (I2 = 69.9%, P = 0.068) using random effect model (Fig. 6).

Figure 6 Forest plot for assessing the association between heavy alcohol consumption and FLD.

Figure 7 Forest plot of subgroup analysis conducted by sex for assessing the association between heavy alcohol consumption and FLD in Japan.

Discussion

Alcohol consumption is a common lifestyle factor and has been associated with cancer, cardiovascular diseases, type 2 diabetes, liver cirrhosis and stroke (Corrao et al., 2004; Ronksley et al., 2011). However, it has been suggested, in contrast, that moderate alcohol consumption shows a beneficial influence on coronary heart disease, stroke, type 2 diabetes mellitus, and cataract (Rimm et al., 1999; Rehm et al., 2003; Koppes et al., 2005; Ronksley et al., 2011; Gong et al., 2015). Similarly, several epidemiological studies have also revealed that moderate alcohol consumption has a protective effect on the development of FLD (Dunn, Xu & Schwimmer, 2008; Gunji et al., 2009; Yamada et al., 2010; Hiramine et al., 2011; Dunn et al., 2012; Hamaguchi et al., 2012; Moriya et al., 2013; Sookoian, Castaño & Pirola, 2014; Hashimoto et al., 2015; Kächele et al., 2015; Moriya et al., 2015; Nishioji et al., 2015; Sogabe et al., 2015) It seems paradoxical because the excessive alcohol consumption causes alcoholic liver diseases (You & Crabb, 2004). As many new epidemiological studies became available, we conducted the separate meta-analysis for the association of LMAC (≤40 g/day or ≤280 g/week) and heavy alcohol consumption (>40 g/day or >280 g/week) with FLD risk.

In the meta-analysis of LM drinkers vs non-drinkers that included the 16 heterogeneous studies (I2 = 79.3%) with 31,942 LM drinkers, we revealed that LMAC was associated with a 22.6% reduction in risk of FLD. The finding was similar to a previous meta-analysis only including 10 heterogeneous studies by Sookoian, Castaño & Pirola (2014), who found that light or modest drinkers (less than 40 g/day of alcohol) had a 31.2% reduction in risk of NAFLD compared with nondrinkers. Then, we conducted an amount-stratified analysis with respect to LMAC. The drinkers were classified into light (≤20 g/day) and moderate (>20–40 g/day) drinkers. The result indicated that a greater protective role for FLD was found in the light drinkers (25.3%) compared with the moderate drinkers (19.6%). Further, we want to know if the beneficial effect is influenced by sex and BMI. Our result showed that the protective effect of LMAC on FLD seemed to be greater in the female drinkers (30.2%) and the drinkers with BMI ≥25 kg/m2(31.3%) compared with the male drinkers (22.6%) and the drinkers with BMI >25 kg/m2(21.3%), respectively. Consistent with our findings, Sookoian et al. also showed that the protective effect of light or modest alcohol consumption on NAFLD was significantly higher in women than in men; but this beneficial effect was not influenced by BMI in the study by Sookoian et al., which was different from our result described above. The differences between our results and the study by Sookoian et al. may be explained by a larger number of included subjects in our meta-analysis. Taken together, the above findings suggest that LMAC may significantly reduce risk of FLD development in our studied population, and especially show a greater protective role for women and obese population.

In the meta-analysis of heavy drinkers vs nondrinkers, the eight heterogeneous studies (I2 = 93.6%) (six conducted in Japan and two in Germany) with 5,468 heavy drinkers were included. The pooled OR showed that heavy alcohol consumption was not statistically associated with risk of FLD. Then, the above eight studies were divided into two groups according to study areas. The pooled result from the six Japanese studies showed a 33.2% reduction in risk of FLD in relation to heavy alcohol consumption. Further subgroup analysis by sex indicated that heavy alcohol consumption had no significant influence on risk of FLD in Japanese women, but yielded a 33.7% reduction in risk of FLD in Japanese men. However, in the meta-analysis from the two Germanic studies that included 132 heavy drinkers and 213 male heavy drinkers, an increased risk of FLD was found in relation to heavy alcohol consumption. Concordant with our results from Japanese studies, Knott et al. reported that reductions in the risk of type 2 diabetes were present at all levels of alcohol intake <63 g/day (Knott, Bell & Britton, 2015); Larsson, Orsini & Wolk (2015) noted that high alcohol consumption (≥14 drinks/week) did not increase risk of heart failure. However, because of the limited studies and small sample sizes, the effect of heavy alcohol consumption on FLD remains unclear, and more prospective studies are needed.

The causal impact of alcohol on liver cirrhosis has long been known. However, the likelihood of developing progressive alcohol-induced liver disease or cirrhosis is not completely dose-dependent, because it occurs in only a subset of patients (O’Shea, Dasarathy & McCullough, 2010). Bellentani et al. (1997) reported that, in a population-based cohort study of almost 7000 subjects in Italy, even among subjects with very high daily alcohol intake (120 g/day), only 13.5% developed ALD, which means that heavy alcohol consumption is likely not to increase risk of liver disease in most of the subjects. It has been shown that the development and progression of alcohol-associated liver disease may depend upon multiple risk factors, including the dose, duration, and type of alcohol consumption, drinking patterns, sex, ethnicity, and genetic factors, and so on (O’Shea, Dasarathy & McCullough, 2010).

In the two previous meta-analysis conducted by Corrao et al. (1998) and Rehm et al. (2010), they included 15 and 17 epidemiological studies, respectively, mainly from the USA and Europe, and assessed the association between alcohol consumption and liver cirrhosis, demonstrating that heavy alcohol consumption significantly increases risk of liver cirrhosis. Corrao et al. also found that the same amount of average alcohol consumption was related to a higher risk of liver cirrhosis in women than in men. In the present meta-analysis, we evaluated the association of alcohol consumption with risk of FLD by including the 16 observational studies mainly from Asia, especially Japan. Our results, which have been described above, are inconsistent with the findings in the two previous meta-analysis. The different results between our study and the two previous meta-analysis may be explained in part by the differences in different stages of FLD development, and ethnicity and genetic factors. The two meta-analysis by Corrao et al. and Rehm et al. assessed the association of alcohol consumption with frank liver cirrhosis, namely end-stage liver disease of ALD development, whereas our meta-analysis evaluated the correlation between alcohol consumption and risk of the relative early stages of ALD development, namely fatty liver (simple steatosis) and steatohepatitis. On the other hand, Kwon et al. reported that aldehyde dehydrogenase 2 (ALDH2) deficiency can ameliorate alcoholic fatty liver in mice (Kwon et al., 2014). Approximately 40–50% of East Asians carry an inactive ALDH2 gene (ALDH2*2 allele) (Singh et al., 1989), but it is very rarely that ALDH2*2 allele is found in European (Peterson, Goldman & Long, 1999). The above data appear to partly explain the reason why even excessive alcohol consumption also seemed to have a protective effect on FLD in Japanese men.

There were limitations to our meta-analysis that should be considered. The main limitation of this study was a small number of included studies and subjects (only 16 studies and 76,608 subjects), so further subgroup analysis were not able to be perform according to type of alcoholic beverages, frequency of alcohol consumption, duration of alcohol consumption, study region and age groups. Secondly, this meta-analysis contained only one cohort study, and the remaining 15 were cross-sectional studies that signify a low quality, because self-reported data on alcohol consumption in epidemiological studies may not be reliable. Thirdly, in the 10 Asian studies included, nine were from Japan and one was from Hong Kong, thus the study coverage in Asian was limited because of absence of studies from other Asian countries, especially Chinese Mainland. Moreover, just 6 studies from other countries (USA, Brazil, Argentina and Germany) were included, thus the study coverage in the world was limited because of absence of studies from Africa and Australia, and a small number of the studies from the USA and Europe. Therefore, the value of our results is limited for other areas except the countries involved in the study (such as China, Africa, Australia, most European countries, and so on). Fourthly, because FLD is a multi-factorial disease, it is uncertain whether other factors may have influenced the results. Fifthly, because early stages of ALD are often asymptomatic, and most of subjects in the included studies were asymptomatic from health check-up at hospital, therefore the results of meta-analysis from these studies can’t be effectively broadened so as to represent the population at large. Lastly, potential publication bias might have influence the results, despite no bias indicated from either the funnel plot or Egger’s test.

Conclusions

In summary, LMAC is associated with a significant protective effect on FLD in the studied population, especially in the women and obese population. However, the effect of heavy alcohol consumption on FLD remains unclear due to limited studies and small sample sizes.

However, because of the accepted involvement of alcohol consumption, especially excessive drinking in liver disease or cirrhosis, these findings should be treated with caution. Further better prospective studies are needed to answer the question of whether alcohol consumption has a diverse effect on FLD in different areas, and whether different kinds of beverages or drinking patterns have a diverse effect on FLD.

Supplemental Information

Data S1 Raw data applied for Tables 1, 2 and 3, and for meta-analyses to generate forest plots of Figures 2 through 7

Click here for additional data file.

Additional Information and Declarations

Competing Interests

Author Contributions

Data Availability

The authors declare there are no competing interests.

Guoli Cao performed the experiments, analyzed the data, wrote the paper, prepared figures and/or tables, reviewed drafts of the paper.

Tingzhuang Yi and Qianqian Liu performed the experiments, analyzed the data, contributed reagents/materials/analysis tools, reviewed drafts of the paper.

Min Wang contributed reagents/materials/analysis tools, reviewed drafts of the paper.

Shaohui Tang conceived and designed the experiments, wrote the paper, prepared figures and/or tables, reviewed drafts of the paper.

The following information was supplied regarding data availability:

The raw data has been supplied as Data S1.

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
