# Peer review of "Alcohol consumption and risk of fatty liver disease: a meta-analysis"

_PeerJ, doi:10.7717/peerj.2633_

## Round 0.1 · original submission · Minor Revisions

· Academic Editor

Minor Revisions

Thank you for your submission. Please address the concerns of the reviewers, it is suggested that you include some further discussion around the limitations of the small sample sizes for specific groups and that you soften the language of your conclusions regarding the "significant predictive effect" to be limited to the specific populations studied.

They have also noted a number of typographical and grammatical errors. The manuscript should be proof read prior to re-submission.

·

Basic reporting

No Comments.

Experimental design

Good.

Validity of the findings

Good.

Additional comments

23 Aug 2016

Comments to the Author

GuoLi Cao et al. performed a systemic review, in which authors focused on the relationship among alcohol consumption and fatty liver disease.

This review manuscript encompasses an interesting topic and may be worthwhile publishing, however it needs a minor revision, scientific and stylistic.

Major comments:
1. In subgroup analysis, authors concluded that heavy alcohol consumption may not be related to risk of FLD in Japanese women, but may have a significant protective role for FLD in Japanese men and conversely may have a detrimental effect on FLD in Germans. The relationships of heavy alcohol consumption in Japanese and German was inconsistent. The number of study was as small as 6 in Japanese study and only 2 in German study. And, it is difficult to eliminate the effect of co-factors except for alcohol consumption. Thus, the effect of heavy alcohol consumption on FLD has been unsettled. Please reconsider the conclusion of this systemic review.

Specific comments:
1. In several sentences, author wrote “sentence no space ( ref)”. Please add space in front of (ref).”

·

Basic reporting

Page 2 line 22 - Consider using “obese populations” or “the obese population” in the abstract conclusion for grammatical clarity.

Page 4 line 16 -NAFLD is considered by most authors to be the hepatic manifestation of the metabolic syndrome so it may be stated as such.

Page 4 line 22 - and even heavier alcohol consumption.

Page 4 line 23 -It should be mentioned that these are all suggested mechanisms of protection, but not all necessarily proven

Page 9 lines 14-16 - should be revised to add clarity

Page 13 line 3 – Cataract formation, the formation of cataracts?

Page 13 line 10 – Requires review of grammar. Also, though it is not inevitable, fatty liver itself is recognised to be a consequence of excessive alcohol consumption, rather than just the late stage manifestations of cirrhosis & cancer.

Page 13 line 13 – Studied should be studies, should this then also be became rather than become?

Page 14 lines 5-10 – Very long run-on sentence that requires additional punctuation for clarity. Should be findings rather than finding. Referencing the same author twice in the sentence should not be necessary.

The introduction is clear & provides a good background to show the context within which the review was performed.

It may be also mentioned as a limitation that early stages of ALD are often asymptomatic. Therefore can the results of meta-analysis from these studies be effectively broadened so as to represent the population at large? Clarify whether the studies examined were searching for FLD in the population or were these patients already manifesting symptoms?

Figures used are relevant & clearly labelled. The raw data was supplied.

Experimental design

The research question is well defined, and the strategy used to undertake this appears robust. It is made clear what the intent of the authors is in undertaking this meta-analysis.

The search strategy was broad enough to capture what I suspect is the vast majority of relevant works, one might ask if “wine” and “beer” were search terms then could “spirits” not have also been relevant? Though this may not necessarily add any further specific results.

The methods are described in such a way as to be easily reproducible. The data analysis appears to have been effectively undertaken.

It may be of interest to state what informed the definition of the four groups of drinkers by levels of alcohol consumption, given that there may be global variation in these standards. Especially as specific population groups are later identified as being potentially protected by them.

Validity of the findings

The limitations of the review are stated, and the further questions that it raises are noted. It is accepted that the findings can not necessarily be accepted purely as seen, and that there is a requirement for further study.

The conclusion is clear & relates well to the original research question, but the abstract gives the appearance of more concrete claims. I would be loath to use the term “likely to have a significant protective effect on the development of FLD…” as the mechanisms are poorly understood & the studied populations so limited. This may be accepted if this were clarified to be only within the studied populations.

---

## Round 0.2 · accepted · Accept

· Academic Editor

Accept

Thank you for your submission. You have addressed all of the reviewers concerns and I am pleased to inform you that your paper is now accepted for publication.

On several occasions you use the term "unsettled" when referring to the effect of heavy alcohol consumption - for clarity I suggest that you change this to " remain unclear" when uploading your final documents